# Effects of ciprofol infusion on hemodynamics during induction and maintenance of anesthesia and on postoperative recovery in patients undergoing thoracoscopic lobectomy: Study protocol for a randomized, controlled trial

**Na Guo**[1,2,3], **Jianqiao Cao**[1,2,3], **Mingjie Duan**[1,2,3], **Fei Zhou**[1,2,3], **Wei Wang**[1,2,3], **Lingling Xu**[4], **Chuansong Wei**[4], **Xiumei Song**[1,2,3]*

1 Department of Anesthesiology, The First Affiliated Hospital of Shandong First Medical University & Shandong Provincial Qianfoshan Hospital, Ji'nan, China, 2 Shandong Institute of Anesthesia and Respiratory Critical Medicine, Ji'nan, China, 3 Shandong Provincial Clinical Research Center for Anesthesiology, Ji'nan, China, 4 Department of Nursing, The First Affiliated Hospital of Shandong First Medical University & Shandong Provincial Qianfoshan Hospital, Ji'nan, China

* ssm801117@163.com

**Data Availability Statement:** No datasets were generated or analysed during the current study

## Abstract

### Introduction

Ciprofol, a new candidate drug, is effective and safe for the maintenance of anesthesia in non-cardiothoracic and non-neurological elective surgery. However, few studies have been conducted on general anesthesia using ciprofol in patients undergoing thoracoscopic lobectomy. Therefore, this study aims to observe the effects of ciprofol on hemodynamics and on postoperative recovery in patients undergoing thoracoscopic lobectomy.

### Methods and analysis

This randomized controlled trial will include 136 patients aged 18–65 years undergoing elective thoracoscopic lobectomy between April 2023 and December 2024. The participants will be randomly assigned to the propofol or ciprofol group. The primary outcome to be assessed is the hemodynamic fluctuation during the induction and maintenance of anesthesia. The secondary outcomes involve quality of anesthesia induction and quality of recovery from anesthesia. The former includes $T_{LOC}$ (time to loss of consciousness), the use of vasoactive agents, the incidence of injection pain, body movement, muscle twitching and coughing during induction of anesthesia. The latter includes $T_{ROC}$ (time to recovery of consciousness), post anesthesia care unit (PACU) time, incidence of postoperative nausea and vomiting (PONV), postoperative agitation, intraoperative awareness and quality of recovery (QoR) score.

now. Data supporting this study will be available from https://figshare.com/ at DOI:10.6084/m9.figshare.25421503, when the clinical trial is completed.

**Funding:** The author(s) received no specific funding for this work.

**Competing interests:** The authors have declared that no competing interests exist.

## Discussion

A number of clinical trials have confirmed that ciprofol, as a new sedative-hypnotic agent, has advantages of better tolerance, higher sedation satisfaction score, and lower incidence of adverse reactions, especially in reducing the incidence of injection pain. But considering that ciprofol was recently developed, limited data are available regarding its use for general anesthesia. This study aims to investigate the effects of ciprofol on hemodynamics and on postoperative recovery of patients undergoing thoracoscopic lobectomy. The results of this study may provide evidence for the safe application of ciprofol, a new choice of general anesthetic for thoracic surgery.

## Clinical trial registration

ClinicalTrials.gov (NCT05664386).

## 1 Introduction

Ciprofol, a new drug candidate for general anesthesia [1, 2], also is known as HSK3486. It was approved for the induction of general anesthesia in December 2020 and anesthesia maintenance in March 2022 by state food and drug administration (SFDA) of China [3]. Similar to propofol, ciprofol is a novel 2,6-disubstituted phenolderivative that binds to γ-aminobutyricacid-α (GABA-A) receptors [4]. Multiple preclinical and clinical studies have confirmed that ciprofol possesses dose-related sedative-hypnotic effects, rapid onset, rapid offset, a potency that is 4–6 times greater than that of propofol, and minor residual side effects following the administration of a single therapeutic dose [4, 5].

Ciprofol is currently used for the induction and maintenance of general anesthesia in adult patients undergoing elective surgery, sedation during endoscopy of the digestive tract and sedation in the ICU. The preliminary study demonstrated that general anesthesia can be successfully induced using 0.2–0.4 mg/kg of ciprofol in elderly patients who underwent major noncardiac surgery without the incidence of any serious adverse events [6–8]. Qin et al. [9] reported that ciprofol could be used safely and effectively to induce and maintain anesthesia in patients who underwent kidney transplantation, and it had superior sedative effects compared with propofol. However, clinical trial participants were patients who underwent non-cardiothoracic and non-neurological elective surgeries.

Due to its newly developed, few studies have been conducted to explore the use of ciprofol in thoracic anesthesia. Thoracic surgery requires a double-lumen endotracheal tube to achieve lung isolation, which needs to pay special attention to the anesthetic management during induction and maintenance of general anesthesia. Propofol is most commonly used for general anesthesia and favored by anesthesiologists due to its advantages of good recovery, short half-life, rapid elimination, small residual sedation, and low incidence of vomiting [10]. However, the administration of propofol is associated with the incidence of hemodynamic instability, especially hypotension [11]. It has been proved that intraoperative hypotension is strongly associated with delayed recovery from anesthesia, myocardial injury, stroke, acute kidney injury, and death. Patients who underwent fiberoptic bronchoscopy used ciprofol for sedation had more stable hemodynamics, lower risk of respiratory depression than those used propofol [12, 13]. Compared with propofol, whether the use of ciprofol for induction and maintenance of general anesthesia can reduce incidence of intraoperative hypotension and accelerate

| | | | STUDY PERIOD | | | | |
|---|---|---|---|---|---|---|---|
| | Enrolment | Allocation | Anesthesia induction | Anesthesia maintenance | PACU | POD1 | POD2 |
| Timepoint | −D 1 | −D 1 | 0 | 0 | 0 | D1 | D2 |
| **Enrolment:** | | | | | | | |
| Eligibility screen | X | | | | | | |
| Informed consent | X | | | | | | |
| Allocation | | X | | | | | |
| **Interventions:** | | | | | | | |
| [Propofol Group] | | | ⟷ | ⟷ | | | |
| [Ciprofol Group] | | | ⟷ | ⟷ | | | |
| **Assessments:** | | | | | | | |
| [Baseline data] | X | | | | | | |
| SBP,DBP,MAP,HR | X | | ⟷ | ⟷ | | | |
| BIS | X | | ⟷ | ⟷ | | | |
| $T_{LOC}$ | | | X | | | | |
| $T_{ROC}$ | | | | | X | | |
| [Extubation time] | | | | | X | | |
| [Ramsay score] | | | | | X | | |
| [SAS score] | | | | | X | | |
| [NRS score] | | | | | X | | |
| [PONV] | | | | | X | X | X |
| [Intraoperative awareness] | | | | | | X | |
| [QOR score] | X | | | | | X | X |
| [PACU time] | | | | | X | | |

**Fig 1. Study timeline and schedule of enrolment, allocation, interventions, and assessments.** SBP: Systolic blood pressure; DBP: Diastolic blood pressure; MAP: mean arterial pressure; HR: heart rate; BIS: bispectral index; $T_{LOC}$: time to loss of consciousness; $T_{ROC}$: time to recovery of consciousness; SAS score: Ricker Sedation-Agitation Scale score; NRS score: Numerical rating scale score; PONV: postoperative nausea and vomiting; QOR score: quality of recovery score; PACU time: length of stay in the post anesthesia care unit.

postoperative recovery in patients undergoing thoracoscopic surgery remains unclear. This study aims to investigate the effects of ciprofol on hemodynamics and on postoperative recovery in patients undergoing thoracoscopic lobectomy.

## 2 Materials and methods

### 2.1 Study design

This randomized controlled trial has been approved by the Ethics Committee of The First Affiliated Hospital of Shandong First Medical University (YXLL-KY-2023(042)) and has been registered at clinicaltrials.gov (NCT05664386). This study protocol is created in accordance with the guidelines of good clinical practice and the Declaration of Helsinki [14, 15]. And it complies with the Standard Protocol Items: Recommendations for Interventional Trials (SPIRIT) guidelines [16]. This trial will be conducted at the First Affiliated Hospital of Shandong First Medical University in Jinan City, Shandong Province, China. A total of 136 participants will be randomly allocated to the propofol group (n = 68) or the ciprofol group (n = 68). The study has been initiated in April 2023, and the expected completion date is December 2024. The study timeline and schedule of enrolment, allocation, interventions, and assessments are shown in **Fig 1**.

The flow diagram of study is shown in **Fig 2**.

### 2.2 Sample size estimation

The primary outcome that will be assessed in this study is the perioperative hemodynamic fluctuation, and mean arterial pressure (MAP) will be used as the evaluation index. We used a Repeated Measures analysis of variance (ANOVA) to calculate the sample size in the PASS 2021(v21.0.3). According to the results of the preliminary experiment, we observed 15 measurements of MAP ($T_{0-6}$ and $OR_{0-7}$) per patient during induction and maintenance of

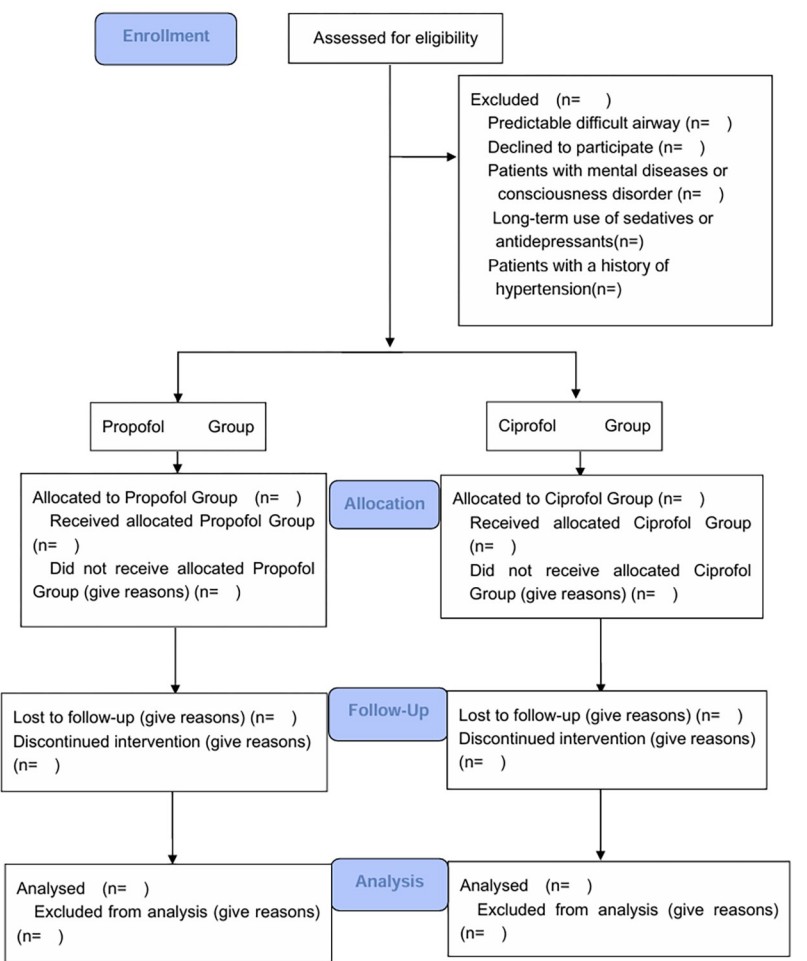

**Fig 2. Flow diagram of study.**

anesthesia between the two groups, the standard deviation of each group is approximately 13, and the autocorrelation coefficient between the adjacent measurement points of the same study participant is 0.5, the difference of the MAP between the two groups is 3. Type I error is 5% and the power is 80% in the pre-test. In the bilateral test, the sample size of the two groups is equal, and the sample size of each group is 54. Considering a 20% drop out rate, a sample size of 68 patients in each group is required. Thus, the final sample size is 136.

## 2.3 Inclusion criteria

Patients who meet the following inclusion criteria will be included in the study: American Society of Anesthesiology (ASA) physical status class I–II; aged between 18 and 65 years; body mass index (BMI) of 20–30 kg/m$^2$; and scheduled to undergo thoracoscopic lobectomy under general anesthesia.

## 2.4 Exclusion criteria

Patients who meet the following one or more exclusion criteria will be excluded from the study: patients or their families refuse to participate in the clinical study; predictable difficult airway; allergy to any of the drugs used in this study; patients with mental illnesses or altered

consciousness; patients with long-term use of sedative-hypnotic/anxiolytics agents; patients with a history of hypertension; patients involved in other clinical studies.

## 2.5 Withdrawal criteria

Withdrawal criteria includes: more than two endotracheal intubation attempts; the operation takes less than 30 minutes; massive intraoperative hemorrhage (blood loss of more than 500mL); accidental injuries from surgery (e.g., surgical procedures cause damage to the heart or large blood vessels, which leads to severe hypotension); perioperative anaphylaxis; conversion of thoracoscopic surgery to thoracotomy; and withdrawal of informed consent or request to withdraw from the study during the observation period. These participants will no longer be included in the analysis.

## 2.6 Study implementation

**2.6.1 Anesthesia and monitoring methods.** All participants will fast for 8 h before the surgery, and water intake will be restricted for 2h before the surgery. Upon arrival in the operating theater, an intravenous catheter will be placed in the fore arm of the patient for intravenous infusion. The participants will be continuously monitored using electrocardiography (ECG), pulse oxygen saturation ($SpO_2$), and the bispectral index (BIS) of electroencephalography (EEG). An intra-arterial cannula will be placed in the radial artery for continuous arterial pressure measurement. End-tidal carbon dioxide concentration ($PetCO_2$) will be monitored after intubation. A 35F or 37F left double-lumen tracheal tube will be selected according to the height and sex of the patient. The participants will be divided into the propofol group (group P) or ciprofol (group C) group according to the allocation.

**2.6.2 Intravenous anesthesia induction.** The induction of general anesthesia is defined as the stage from the injection of ciprofol or propofol to 5 minutes after the completion of tracheal intubation. The patients will recieve continuous infusion of 0.9% sodium chloride at a rate of 6–8 mL/kg/h, and be preoxygenated with 100% oxygen at 6 L/min through a face mask during the induction stage. The patients in group P will receive 2–2.5mg/kg of propofol for induction, whereas those in group C will receive 0.4–0.5 mg/kg of ciprofol for induction. If the BIS value is ≤ 60, 0.2 mg/kg of cisatracurium and 0.5μg/kg of sufentanil will be administered, and double-lumen endotracheal tube intubation will be performed after skeletal muscles have relaxed. If the BIS value is > 60, 1 mg/kg of propofol or 0.2 mg/kg of ciprofol will be immediately added, with an interval of > 1min between each bolus, until the BIS value is ≤ 60, 0.2 mg/kg of cisatracurium and 0.5 μg/kg of sufentanil will be administered, and intubation will be performed. If ciprofol or propofol is added more than three times, the BIS value is still > 60, and the induction will be deemed to be failed. Midazolam 0.05–0.15 mg/kg will be alternative to complete the induction. The continuous infusion of ciprofol or propofol will be initiated after induction. A fibrobronchoscope will be used for positioning double-lumen tube, then the patient will be placed in the lateral decubitus position and the double-lumen tube will be positioned again. Volume-controlled ventilation will be performed (tidal volume of two-lung ventilation 6–8 mL/kg, one-lung ventilation 4-6mL/kg, respiratory rate, 10–18 times/min) and $PetCO_2$ will be maintained at 30–40 mmHg. Blood pressure and heart rate (HR) fluctuation should be maintained at no more than 30% of the basal value during the induction stage (The basal values for blood pressure and HR are defined as the average values of three cuff blood pressure and HR at different time points after hospitalization), vasoactive drugs could be used when necessary, and the dosage and frequency of vasoactive drugs will be recorded.

**2.6.3 Anesthesia maintenance.** Maintenance is defined as the time from end of induction to the discontinuation of anesthetics (The infusion of anesthetic agents will be stopped immediately at the end of surgery). Group P will receive continuous propofol infusion at a rate of 4–12 mg/kg/h for maintenance, whereas Group C will receive continuous ciprofol infusion at a rate of 0.4–2.4 mg/kg/h for maintenance. Remifentanil will be administered at a rate of 0.1–0.3 μg/kg/min, and cisatracurium and sufentanil will be added in both groups according to requirements of the surgery. One-lung ventilation will be initiated at the beginning of the surgery, and the respiratory parameters will be adjusted to maintain a $PetCO_2$ of 30–40 mmHg. The intraoperative BIS value will be maintained at 40–60. Vasoactive agents will be used when blood pressure and/or HR fluctuation more than 30% of the basal value under adequate depth of anesthesia. All anesthetics administration will be discontinued when the surgical procedures are completed. The patients will be provided with patient-controlled intravenous analgesia (PCIA) (2 μg/kg of sufentanil + 2 mg/kg of flurbiprofen axetil + 24 mg of ondansetron, total volume 100 mL, 2 mL/h for 48 h). The self-controlled capacity and the locking time will be 0.5 mL and 15 minutes, respectively.

**2.6.4 Recovery of anesthesia.** All patients will be transferred to the PACU after the surgery and received standard monitoring (ECG, SpO2, NIBP). Tracheal extubation will be performed after confirming the recovery of consciousness (ROC defined as responses to verbal commands), and sufficient spontaneous breathing. When Steward awakening score is ≥ 4 points, the patients will be allowed to return to the ward and the length of stay in the PACU will be at least 30 minutes after removing endotracheal tube.

The Ricker Sedation-Agitation Scale (SAS) [17] and the Ramsay score at extubation will be evaluated immediately. The following events will be recorded during this period: $T_{ROC}$, time to recovery of spontaneous breathing, extubation time, numeric rating scale (NRS) score for pain [18] after recovery, the incidence of PONV, and the length of PACU stay. The quality of recovery (QOR)-15 questionnaire [19] will be used once before the surgery and then on postoperative day 1, 2 (POD1, POD2) to assess the quality of recovery. The Brice questionnaire [20, 21] will be used to assess intraoperative awareness on POD1. The protocols are detailed in **Table 1**.

## 2.7 Outcome

**2.7.1 Primary outcomes.** The primary outcomes are MAP and HR at following time points during induction and maintenance.

MAP and HR will be recorded during the induction stage, including the following time points: before induction ($T_0$), after induction ($T_1$ BIS≤60), immediately before intubation ($T_2$), immediately after intubation($T_3$), 1, 3, and 5 minutes after intubation($T_4$, $T_5$, $T_6$).

MAP and HR will be recorded during anesthesia maintenance, including the following time points: when the patient is placed in the lateral decubitus position ($OR_0$), when fiberoptic bronchoscope using for localization at the second time ($OR_1$), immediately before the surgery ($OR_2$), 1, 3, 5, and 30 minutes after surgical incision and at the end of surgery ($OR_3$, $OR_4$, $OR_5$, $OR_6$, $OR_7$).

**2.7.2 Secondary outcomes.** The secondary outcomes include: quality of anesthesia induction and quality of reco-very from anesthesia.

Quality of anesthesia induction includes: $T_{LOC}$ is defined as the duration from ciprofol or propofol administration to achieving a BIS of≤60; the use of vasoactive agents, the incidence of injection pain, body movement, muscle twitching and coughing.

Quality of recovery from anesthesia includes: $T_{ROC}$ is defined as the duration from the discontinuation of all anesthetics to response to verbal commands and sufficient spontaneous

**Table 1. Detailed interventional protocols in the propofol group and ciprofol group.**

| Propofol Group | Ciprofol Group |
|---|---|
| **Before surgery** | |
| monitoring methods:• SpO$_2$<br>• ECG<br>• PetCO$_2$<br>• BIS<br>• invasive blood pressure | |
| **Anesthesia induction** | |
| • Propofol 2–2.5mg/kg | • Ciprofol 0.4–0.5mg/kg |
| • If BIS ≤ 60, cisatracurium0.2mg/kg and sufentanil 0.5μg/kg are injected to finish induction | • If BIS ≤ 60, cisatracurium 0.2mg/kg and sufentanil 0.5μg/kg are injected to finish induction |
| • If BIS > 60, add propofol (1mg/kg, each time), interval > 1min, less than 3 times,until BIS ≤ 60, add cisatracurium and sufentanil to finish induction; if BIS still > 60, induction fails, midazolam will be alternative to finish induction | • If BIS > 60, add ciprofol (0.2mg/kg, each time), interval > 1min, less than 3 times,until BIS ≤ 60, add cisatracurium and sufentanil to finish induction; if BIS still > 60, induction fails, midazolam will be alternative to finish induction |
| **Anesthesia maintenance** | |
| Propofol 4-12mg/kg/h | Ciprofol 0.4–2.4mg /kg/h |
| Remifentanil 0.1–0.3ug/kg/min | Remifentanil 0.1–0.3ug/kg/min |
| Add sufentanil and cisatracurium according to the requirements of the surgery. | Add sufentanil and cisatracurium according to the requirements of the surgery. |
| BIS 40–60 | BIS 40–60 |
| **Post Anesthesia Care Unit (PACU)** | |
| • Ramsay score<br>• Ricker Sedation-Agitation Scale (SAS)<br>• Numeric Rating Scale (NRS) for pain<br>• Postoperative nausea and vomiting (PONV)<br>• Brice questionaire<br>• Time to recovery of consciousness<br>• Time to recovery of spontaneous breathing<br>• Extubation time<br>• PACU time | |

breathing. PACU time is defined as the time from entering the PACU to exiting the PACU. Incidence of PONV, postoperative agitation, intraoperative awareness and QoR score.

The QoR-15 questionnaire is a patient-reported outcome measure that has been validated to measure QoR after general anesthesia. Its scores range from 0 to 150, with higher scores indicating better recovery [22]. Based on the scores, QoR can be classified into excellent (QoR-15 >135), good (122≤QoR-15 ≤135), moderate (90 ≤ QoR-15 ≤ 121),and poor (QoR-15 <90).The patients will complete the QoR-15 questionnaire at three time points: the day before the surgery, POD1 and POD2 [23].

## 2.8 Adverse events

**2.8.1 Tachycardia.** Tachycardia is defined as a HR > 100 beats/min or an increase of more than 30% from baseline if the baseline value is > 78 beats/min. Patients with tachycardia will receive 10 mg of esmolol, or the anesthetic dose will be adjusted.

**2.8.2 Bradycardia.** Bradycardia is defined as a HR < 50 beats/min or a reduction of more than 30% from the baseline if the baseline value is < 71 beats/min. Patients with bradycardia will receive 0.5 mg of atropine or 2 μg of isoproterenol, or the anesthetics dose will be adjusted.

**2.8.3 Hypertension.** Hypertension is defined as an increase in MAP 30% of baseline. Patients with hypertension will receive 10 mg of urapidil, or the anesthetic dose will be adjusted.

**2.8.4 Hypotension.** Hypotension is defined as a decrease in MAP 30% of baseline. Patients with hypotension will receive a liquid infusion, 6 mg of ephedrine, or 4 μg of norepinephrine, or the anesthetic dose will be adjusted.

**2.8.5 Intraoperative awareness.** Patients with intraoperative awareness can recall intraoperative events that occurred under general anesthesia, including pain, paralysis, or feelings of impending death.

## 2.9 Informed consent

The researchers will obtain a signed informed consent form from each participant before the surgery and are responsible for explaining the purpose, methods, expected benefits, and potential risks of the clinical trial to the study participants. If new safety studies indicate significant changes in the benefit-risk assessment, informed consent will be reviewed and modified, and all study participants will be provided access to new information. Participants will receive an amended informed consent form and the researchers will obtain their informed consent to continue with the study. Study participants will be screened according to the inclusion and exclusion criteria after signing the informed consent form.

## 2.10 Randomization and blinding

Randomization will be performed by an independent researcher, using generated random number after the assessment of eligibility.

1. Patients will be numbered as 1, 2, 3. . . . . . 136, according to the order of enrolment.

2. Random numbers will be obtained for each patient, starting from any number in the random number table, according to the entry sequence in the same direction.

3. Participants with odd numbers will be assigned to the propofol group and those with even numbers will be assigned to the ciprofol group.

4. Adjustment: If one arm reaches 68 patients, the remaining patients will continue to receive random numbers until both groups reach at least 68 cases.

5. The generated random allocation will be placed in an opaque sealed bag in advance and handed over to the nurse who is responsible for preparing medications. If the patient meets the inclusion and exclusion criteria and has signed the informed consent, the anesthesiologist of the patient and the nurse will open the opaque sealed bag sequentially on the day of surgery and the consenting patient is treated according to allocation. The patients will be blinded to the group allocation in this study. Another blinded independent researcher will be responsible for preoperative visiting and obtaining informed consent with participants. The outcome assessors and the data analysts will also be blinded to the allocation.

## 2.11 Data management

Data will be collected by independent researchers using Case Report Form (CRF). If there is a missing value, we will use the method of multiple imputation for processing and conduct sensitivity analysis. All researchers should follow the standard operation procedure (SOP)and study protocol rigorously. Data should be recorded in a timely, direct, accurate, clear, signed, and dated manner. The accuracy and completeness of the data records must be evaluated, and

the errors must be corrected in accordance with the prescribed methods. Verified and reliable statistical software will be adopted for the statistical processing of the data, and effective quality control measures will be adopted for data input, such as double or double entries (lose-lose method and double entry). Investigators will conduct interim analysis to make informed decisions about whether to continue the trial or revise the study design to optimize the trial outcomes.

## 2.12 Statistical analysis

Repetitive measure analysis of variance (ANOVA) and Generalized estimating equations (GEE)will be employed to analyze the repeated measurement data, including MAP and HR. Repetitive measure analysis of variance (ANOVA) will be employed to analyze the QoR-15 score. The measurement data that conform to a normal distribution will be expressed as the mean ± standard deviation ($\bar{x}$±s). Non-normally distributed measurement data are expressed using the median and the $25^{th}$ and $75^{th}$percentiles. Numerical variables (e.g., NRS score for pain, SAS score, Ramsay score, $T_{LOC}$, $T_{ROC}$, extubation time, and PACU time) will be compared between the two groups using the independent t-test or Wilcoxon rank-sum test. Categorical variables (e.g., injection pain, muscle twitching, cough, postoperative agitation, nausea and vomiting, and intraoperative awareness) will be presented as counts(percentages) and compared using χ2 analysis or Fisher's exact test. The overall significance level is set at $p < 0.05$.

## 2.13 Ethical considerations and declarations

Our research is conducted in accordance with the principles of the Declaration of Helsinki (64th WMA General Assembly, October 2013) and was approved by the Ethics Committee of First Affiliated Hospital of Shandong First Medical University (YXLL-KY-2023(042)). Written informed consent will be obtained from all participants and/or their legal representatives. The investigator must report all Serious Adverse Events (SAEs) to the sponsor (the corresponding author in the trial) immediately and inform the Ethics Committee accordingly. The sponsor is responsible for ongoing safety evaluation of the investigational drug (ciprofol and propofol), terminating the trial in the event of SAEs and compensating the participant in case of any SAE.

## 3 Discussion

Thoracoscopic lobectomy often requires the double-lumen endotracheal tube to achieve lung isolation. Due to the larger size and more complex design than the single lumen tube, intubation with a double-lumen tube will have a large impact on hemodynamics, which often required deep sedation and analgesia [24]. Intraoperative hypotension and hypoxemia are common because of cardiac compression, one lung ventilation and restricted fluid intake. During the anesthesia recovery period, patients often have poor breathing due to lobectomy and pain stimulation. Perioperative hypoxemia and hemodynamic instability will lead to poor quality of recovery, be high risk factors for major adverse cardiovascular and cerebrovascular events. General anesthetics used in patients undergoing thoracoscopic lobectomy are well related with intraoperative hemodynamic fluctuation and postoperative quality of recovery. Therefore, the choice of general anesthetics is particularly important. Multiple clinical trials have confirmed that compared with propofol, ciprofol is not inferior in sedative, but less effect on respiratory and circulation, meanwhile reducing injection pain [6–9, 25]. It is not yet known whether ciprofol can relieve the stress response effectively caused by surgery, provide sufficient sedation and maintain hemodynamic stability during thoracoscopic lobectomy.

There is no definite conclusion about the effect of ciprofol on recovery time, also the effect on quality of recovery is still unknown [26, 27]. The influence of ciprofol upon postoperative recovery from anesthesia in patients undergoing thoracic surgery is more complex and requires further study.

Propofol is the 'gold standard' intravenous anesthetic and is commonly used as it has a relatively fast onset, short duration of action, and less residual. However, propofol also has several limitations, such as a narrow therapeutic window, high incidence of hypotension, respiratory depression, and pain upon injection [28]. Median effective dose ($ED_{50}$) of ciprofol is 1.5mg/kg, median lethal dose ($LD_{50}$) is 9.9mg/kg, the therapeutic index (TI) is 6.6, which is about 2.4 times that of propofol, and the safety range is wider. According to previous clinical trials, the incidence of bradycardia, and QT interval prolongation with the use of ciprofol was similar with that of propofol. Moreover, because less effect on circulation and respiratory, less injection pain, ciprofol may be a promising alternative general anesthetic to propofol [7, 12, 13, 29]. We conduct the trial to investigate the effects of ciprofol on perioperative hemodynamics and on quality of recovery from anesthesia in patients undergoing thoracoscopic lobectomy.

However, this study has some limitations. A major limitation of this study will be the small sample size, especially in terms of interpreting the multiple outcomes that this study aims to analyze. Second, the study is not double-blinded, which may introduce bias. Nevertheless, these results may provide evidence for the safe application of ciprofol in thoracic surgery. It is reasonable to expect that ciprofol will become a novel alternative for clinical intravenous anesthesia and bring benefits to patients.

## Supporting information

**S1 Checklist. SPIRIT 2013 checklist: Recommended items to address in a clinical trial protocol and related documents.**
(DOC)

**S1 File. Ethical approval letter.**
(DOCX)

**S2 File. Protocol approved by ethics.**
(DOCX)

## Acknowledgments

We thank the staff at the Center for Big Data Research in Health and Medicine, the First Affiliated Hospital of Shandong First Medical University & Shandong Provincial Qianfoshan Hospital, for their valuable contributions.

## Author Contributions

**Conceptualization:** Mingjie Duan.

**Data curation:** Jianqiao Cao.

**Formal analysis:** Jianqiao Cao.

**Investigation:** Mingjie Duan.

**Methodology:** Fei Zhou, Wei Wang.

**Project administration:** Fei Zhou, Wei Wang.

**Resources:** Lingling Xu, Chuansong Wei.

**Software:** Wei Wang.

**Validation:** Lingling Xu.

**Visualization:** Chuansong Wei.

**Writing – original draft:** Na Guo.

**Writing – review & editing:** Na Guo, Xiumei Song.

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
