## [Decision Letter · Decision Letter 0]

4 Jan 2024

PONE-D-23-32710Effects of Ciprofol Infusion on Hemodynamics during Induction and Maintenance of anaesthesia and on Postoperative Recovery in Patients Undergoing Thoracoscopic Lobectomy: Study protocol for a Randomized, Controlled Trial.PLOS ONE

Dear Dr. Song,

Thank you for submitting your manuscript to PLOS ONE. After careful consideration, we feel that it has merit but does not fully meet PLOS ONE’s publication criteria as it currently stands. Therefore, we invite you to submit a revised version of the manuscript that addresses the points raised during the review process.

**ACADEMIC EDITOR: **please carefully assess all the reviewers comments

We look forward to receiving your revised manuscript.

Kind regards,

Silvia Fiorelli

Academic Editor

PLOS ONE

2. In the online submission form you indicate that your data is not available for proprietary reasons and have provided a contact point for accessing this data. Please note that your current contact point is a co-author on this manuscript. According to our Data Policy, the contact point must not be an author on the manuscript and must be an institutional contact, ideally not an individual. Please revise your data statement to a non-author institutional point of contact, such as a data access or ethics committee, and send this to us via return email. Please also include contact information for the third party organization, and please include the full citation of where the data can be found.

3. PLOS requires an ORCID iD for the corresponding author in Editorial Manager on papers submitted after December 6th, 2016. Please ensure that you have an ORCID iD and that it is validated in Editorial Manager. To do this, go to ‘Update my Information’ (in the upper left-hand corner of the main menu), and click on the Fetch/Validate link next to the ORCID field. This will take you to the ORCID site and allow you to create a new iD or authenticate a pre-existing iD in Editorial Manager. Please see the following video for instructions on linking an ORCID iD to your Editorial Manager account: https://www.youtube.com/watch?v=_xcclfuvtxQ.

5. Please include your table as part of your main manuscript and remove the individual file. Please note that supplementary tables should remain as separate "supporting information" files".

Reviewers' comments:

Reviewer's Responses to Questions

**Comments to the Author**

1. Does the manuscript provide a valid rationale for the proposed study, with clearly identified and justified research questions?

Reviewer #1: Yes

Reviewer #2: No

2. Is the protocol technically sound and planned in a manner that will lead to a meaningful outcome and allow testing the stated hypotheses?

Reviewer #1: Yes

Reviewer #2: No

3. Is the methodology feasible and described in sufficient detail to allow the work to be replicable?

Reviewer #1: Yes

Reviewer #2: No

4. Have the authors described where all data underlying the findings will be made available when the study is complete?

Reviewer #1: Yes

Reviewer #2: No

5. Is the manuscript presented in an intelligible fashion and written in standard English?

Reviewer #1: Yes

Reviewer #2: No

6. Review Comments to the Author

You may also provide optional suggestions and comments to authors that they might find helpful in planning their study.

Reviewer #1: This is a well-designed study, testing a new drug, currently approved for use in China.

My suggestions for improvement:

- If the study is a trial protocol - it should be published before the start of the study, from this protocol in certain places one gets the impression that the study has already been completed, e.g.

- The length of stay in the PACU was at least 30 minutes

- The authors declare that the research was conducted...

- The patients/participants provided their written informed consent...

It is necessary to make it uniform, please specify.

LN. 313 2.10 Sampling - replace this word with "Randomization", because Sampling usually refers to taking samples (laboratory, microbiology, etc.)

According to the protocol for the RCT statement in Supplemental file S1:

- 21b Description of any interim analyzes and stopping guidelines, including who will have access to these interim results and make the final decision to terminate the trial NA

This statement is a necessary part of ethical standards. Data on the safety of ciprofol exist in other categories of patients. However, the possibility of serious side effects related to the use in a special category of patients - thoracic surgery. The authors justified the conduct of this study with a new category of patients. According to the ethical standards - each study manager must terminate the trial in the event of serious adverse events and inform the Ethics Committee accordingly. I propose to include that statement in the Ethical standards.

Given that the Discussion is rather unclear, I suggest that the authors structure it in such a way that they first discuss the possible gains of these studies in relation to the tested drug - ciprofol, then how the comparative drug - propofol - performed in that area, and in which segment ciprofol could be better . If the expected effect is equal to the hemodynamic effect, what would be the benefit of introducing a new drug as an alternative or standard method. Is it the price, faster elimination or something else.

The manuscript has a number of minor grammatical errors, such as capitalizing a noun within a sentence throughout the text. Additional proofreading is therefore required.

Reviewer #2: This protocol is for a randomised controlled trial of ciprofol vs propofol in thorascopic lobectomy. The tense of the protocol, especially the abstract, varies, and could usefully be tidied up.

We are told there are few studies in this area - but none are given in the motivation - it is important here to know that the research is needed and is not merely a means to gaining more experience with ciprofol. To ethically perform a randomised study there needs to be sufficient uncertainty to allocate to either arm and this uncertainty is not expressed in the protocol.

I cannot arrive at the same size given using the data given here. For a single measurement of difference with mean 2.48 and standard deviation 9 this is a standardised difference of about 0.28 - assuming bilateral manes a two-sided test then this requires over 200 patients per group. A sample size of 50 per group is equivalent to a standardised difference of about 5/9 - this calculation needs to be made clear and the method for dealing with apparently repeated measures given here as well.

Terminology needs to be clear - I think shedding criteria are withdrawal criteria but there is a big difference in the reasons here in terms of interpretation - how is this going to be allowed for in the ITT analysis?

See alos 2.10(3) for lack of clarity.

Line 258 implies the list of datapoints is incomplete and the primary outcome is not defined here.

Intergroup and between group are the same thing so there are three tests here and it is not remotely clear what is being done when to which variable. The method of using Bonferroni correction is entirely unclear here - what is the multiple testing?

7. PLOS authors have the option to publish the peer review history of their article (what does this mean?). If published, this will include your full peer review and any attached files.

Reviewer #1: **Yes: **Slavica Kvolik

Reviewer #2: No

---

## [Author Response · Author response to Decision Letter 0]

7 Feb 2024

Dear Editor and Reviewers:

Thank you for your letter and for the reviewers’ comments concerning our manuscript entitled “Effects of ciprofol infusion on hemodynamics during induction and maintenance of anaesthesia and on postoperative recovery in patients undergoing thoracoscopic lobectomy: study protocol for a randomized, controlled trial”.

(Manuscript ID: PONE-D-23-32710).

Those comments are all valuable and very helpful for revising and improving our paper, as well as the important guiding significance to our researches. We have studied comments carefully and have made correction which we hope meet with approval. The reviewers’ comments are laid out below in italicized font and specific concerns have been numbered. Our response is given in normal font.

Revised portion are marked in red in the paper. 

Reviewer 1

 1. Comment: Reviewer #1: This is a well-designed study, testing a new drug, currently approved for use in China.

My suggestions for improvement:

- If the study is a trial protocol - it should be published before the start of the study, from this protocol in certain places one gets the impression that the study has already been completed, e.g.

- The length of stay in the PACU was at least 30 minutes

- The authors declare that the research was conducted...

- The patients/participants provided their written informed consent...

It is necessary to make it uniform, please specify.

 Response: Thank you for your rigorous consideration. We need to explain that due to several reasons, the planned completion date of this study is delayed to December 2024. We have revised the “The study has been initiated in April 2023, and the expected completion date is April 2024.” to “The study has been initiated in April 2023 and the expected completion date is December 2024.”(Page 6 Line 104).

---We have changed this sentence “The length of stay in the PACU was at least 30 minutes” to “The length of stay in the PACU will be at least 30 minutes after removing endotracheal tubes.” (Page 10, Line 204 2.6..4 Recovery of anesthesia ) 

 The revised portion are marked in red.

------We have revised this “ The authors declare that the research was conducted” to “The authors declare that the research will be conducted”, marked in red. (page 2 ,Line 23 Conflict of interest)

----“The patients/participants provided their written informed consent... ”this sentence was originally written in that paragraph “Line 475 Ethics statement ”. According to format requirements of the journal, we deleted this part . We have rewritten the part “Ethical considerations and declarations” in page 17 line 333-341, 2.13 Ethical considerations and declarations. We are very sorry for our mistake. We have revised the tense.

All the revised portion are marked in red in the paper. We gratefully appreciate for your valuable suggestion.

 2.Comment: LN. 313 2.10 Sampling - replace this word with "Randomization", because Sampling usually refers to taking samples (laboratory, microbiology, etc.)

Response: We gratefully appreciate for your valuable comment. We have made correction according to the Reviewer’s comments. ----page 15, line 288 2.10 we replaced “Sampling” with “Randomization and blinding”. Thank you so much for your careful check.

 3.Comment: According to the protocol for the RCT statement in Supplemental file S1:

- 21b Description of any interim analyzes and stopping guidelines, including who will have access to these interim results and make the final decision to terminate the trial NA

This statement is a necessary part of ethical standards. Data on the safety of ciprofol exist in other categories of patients. However, the possibility of serious side effects related to the use in a special category of patients - thoracic surgery. The authors justified the conduct of this study with a new category of patients. According to the ethical standards - each study manager must terminate the trial in the event of serious adverse events and inform the Ethics Committee accordingly. I propose to include that statement in the Ethical standards.

 Response: Thank you for your rigorous consideration. We have revised the text to address your concerns and hope that it is now clearer. In “2.13 Ethical considerations and declarations” paragraph, we have included the statement “The investigator must report all Serious Adverse Events(SAEs) to the sponsor (the correspoding author in this trial ) immediately and inform the Ethics Committee accordingly. The sponsor is responsible for ongoing safety evaluation of the investigational drug (ciprofol and propofol), terminating the trial in the event of SAEs and compensating the participant in case of any SAE. ” (Page 17-18, Line 334-343 ) 

In “ 2.11 Data management ”, we have added this paragraph “Investigators will conduct interim analyse to make informed decisions about whether to continue the trial or revise the study design to optimize the trial outcomes.” (Page16, Line 317-318)

---We revised the Supplemental file S1 “ 21b Description of any interim analyzes and stopping guidelines, including who will have access to these interim results and make the final decision to terminate the trial. NA ” to “ 21b Description of any interim analyzes and stopping guidelines, including who will have access to these interim results and make the final decision to terminate the trial. 16-18 ” 

 4.Comment: Given that the Discussion is rather unclear, I suggest that the authors structure it in such a way that they first discuss the possible gains of these studies in relation to the tested drug - ciprofol, then how the comparative drug - propofol - performed in that area, and in which segment ciprofol could be better . If the expected effect is equal to the hemodynamic effect, what would be the benefit of introducing a new drug as an alternative or standard method. Is it the price, faster elimination or something else.

The manuscript has a number of minor grammatical errors, such as capitalizing a noun within a sentence throughout the text. Additional proofreading is therefore required.

 Response: Thank you so much for your careful check and nice suggestion. 

------We've made extensive changes to this section in manuscript according the reviewer’s comments(First discuss the possible gains of these studies in relation to ciprofol, then how the propofol performed , next, which part the ciprofol could be better than propofol ), please see page18-20，Line 345-381. We appreciate for reviewers’ warm work earnestly, and hope that the correction will meet with approval. 

----We are very sorry for incorrect writing. We have carefully corrected some grammatical errors. 

Other changes: page 9, Line 182 , “The infusion of anesthetic agents will be stop immediately after the surgical procedures” were changed as “The infusion of anesthetic agents will be stopped immediately at the end of surgery.”

-----And here we did not list the changes but marked in red in revised paper.

Once again, thank you very much for your comments and suggestions.

Reviewer 2

 1.Comment: Reviewer #2: This protocol is for a randomised controlled trial of ciprofol vs propofol in thorascopic lobectomy. The tense of the protocol, especially the abstract, varies, and could usefully be tidied up.

Response: We are very sorry for our incorrect writing. 

--------In Abstact part, page 2, line 33-34 “Therefore, this study aimed to observe the effects of ciprofol on hemodynamics and on postoperative recovery in patients undergoing thoracoscopic lobectomy.” were changed as “Therefore, this study aims to observe the effects of ciprofol on hemodynamics and on postoperative recovery in patients undergoing thoracoscopic lobectomy.” 

Other changes were marked in red in revised paper.

 2.Comment: We are told there are few studies in this area - but none are given in the motivation - it is important here to know that the research is needed and is not merely a means to gaining more experience with ciprofol.

Response: Thank you for your rigorous consideration. 

We have revised abstract ,introduction and discussion section carefully. Ciprofol ,a promising intravenous anesthetic drug,is newly developed, has advantages of better tolerance, higher sedation satisfaction score, and lower incidence of adverse reactions, especially in reducing the incidence of injection pain. Some clinical trials also demonstrate that ciprofol has less effect on respiratory and circulation compared with propofol in patients undergoing noncardiacthoracic surgery, we aim to investigate the effect of ciprofol on hemodynamics and postoperative recovery.

Please refer to the revised manuscript file with tracked changes.

----We appreciate for Reviewers’ warm work earnestly, and hope that the correction will meet with approval.

 3. Comment: ---- To ethically perform a randomised study there needs to be sufficient uncertainty to allocate to either arm and this uncertainty is not expressed in the protocol.

Response: We totally understand the reviewer’s concern.

 ------In the part of 2.10 Randomization and blinding we have written that “ The generated random allocation will be placed in an opaque sealed bag and handed over to the nurse in the drug preparation room who prepares the corresponding drug according to the group.” 

------We added the contents about blinding, “The patients will be blinded to the group allocation in this study, whereas, anesthesia providers who will be unblinded to facilitate intraoperative anesthesia management. Another blinded independent researcher will be responsible for preoperative visit and obtaining informed consent with participants. The outcome assessors and the data analysts will also be blinded to the allocation .”（Page 16,line 302-306）. 

 4.Comment: I cannot arrive at the same size given using the data given here. For a single measurement of difference with mean 2.48 and standard deviation 9 this is a standardised difference of about 0.28 - assuming bilateral manes a two-sided test then this requires over 200 patients per group. A sample size of 50 per group is equivalent to a standardised difference of about 5/9 - this calculation needs to be made clear and the method for dealing with apparently repeated measures given here as well.

Response: We feel sorry for the inconvenience brought to the reviewer. For this part of 2.2 Sample size estimation, Page 6, line 112-113, we added this description “We use a repeated measures analysis of variance (ANOVA) to calculate the sample size in the PASS 2021 (v21.0.3)” 

The computational process as follows :

 5.Comment: Terminology needs to be clear - I think shedding criteria are withdrawal criteria but there is a big difference in the reasons here in terms of interpretation - how is this going to be allowed for in the ITT analysis?

Response: We totally understand the reviewer’s concern. We have redefined the appropriate inclusion and exclusion criteria, withdrawal criteria. Intention-to-treat (ITT) analysis aims to include all participants randomized into a trial irrespective of what happened subsequently. We perform withdrawal criteria for patients who meet the inclusion and exclusion criteria ,but withdraw from the trial because of personal events or unpredictable events, including tracheal intubation failed twice or more，intraoperative massive bleeding, anaphylactic shock, or request to withdraw from the study,and so on. We also added this sentence “These participants will no longer be included in the analysis.” We hope that our corrections and explanations will meet with approval. (page 7,line121-140)

 6.Comment: See alos 2.10(3) for lack of clarity.

Response: We feel sorry for the inconvenience brought to the reviewer. We have changed “The remainder of the random number divided by the number of groups can be evenly divided into the propofol group but cannot be evenly divided into the ciprofol group.” to “ Participants with odd numbers will be assigned to the propofol group and those with even numbers will be assigned to the ciprofol group. ”(Page 15,line294-295) We hope our revised content can be clearly expressed.

 7.Comment: Line 258 implies the list of datapoints is incomplete and the primary outcome is not defined here.

Response: We are very sorry for not writing clearly. We have changed “Hemodynamic changes during anesthesia induction and maintenance.” to “The primary outcomes are mean arterial pressure (MAP) and heart rate (HR) at following time points during anesthesia induction and maintenance.” (page 12，line 233). We have added the time point---at the end of surgery(OR10) to ensure the completeness of datapoints.

 8.Comment: Intergroup and between group are the same thing so there are three tests here and it is not remotely clear what is being done when to which variable. The method of using Bonferroni correction is entirely unclear here - what is the multiple testing?

Response: We apologize for our errors. We thank the reviewer for pointing this out. 

-------We have revised as follows: “Repetitive measure analysis of variance (ANOVA) and Generalized estimating equations (GEE)will be employed to analyze the repeated measurement data, including blood pressure and heart rate.Repetitive measure analysis of variance (ANOVA) will be employed to analyze the QoR-15 scores. The measurement data that conform to a normal distribution are expressed as the mean ± standard deviation (x̅±s).Non-normally distributed measurement data are expressed using the median and the 25th and 75thpercentiles. Numerical variables (e.g., NRS score for pain, SAS score, Ramsay score,TLOC, TROC, extubation time, and PACU time) will be compared between the two groups using the independent t-test or Wilcoxon rank-sum test. Categorical variables (e.g., injection pain, muscle twitching, cough, postoperative agitation, nausea and vomiting, and intraoperative awareness)will be presented as counts(percentages) and compared using χ2 analysis or Fisher’s exact test. The overall significance level was set at p < 0.05.” (page 16-17, line319-332)

-----We delete the wrong description , such as “An independent t-test or one-way ANOVA will be used for intergroup comparisons. The Mann–Whitney U test will be used for comparison between the groups”. “and Bonferroni correction will be used to control for type I errors” . 

We have studied comments carefully and have made correction which we hope meet with approval.

We would like to thank the reviewers again for taking the time to review our manuscript.

---

## [Decision Letter · Decision Letter 1]

5 Mar 2024

PONE-D-23-32710R1Effects of Ciprofol Infusion on Hemodynamics during Induction and Maintenance of anaesthesia and on Postoperative Recovery in Patients Undergoing Thoracoscopic Lobectomy: Study protocol for a Randomized, Controlled Trial.PLOS ONE

Dear Dr. Song,

Thank you for submitting your manuscript to PLOS ONE. After careful consideration, we feel that it has merit but does not fully meet PLOS ONE’s publication criteria as it currently stands. Therefore, we invite you to submit a revised version of the manuscript that addresses the points raised during the review process.

**ACADEMIC EDITOR: **please carefully assess all the reviewers comments

We look forward to receiving your revised manuscript.

Kind regards,

Silvia Fiorelli

Academic Editor

PLOS ONE

Reviewers' comments:

Reviewer's Responses to Questions

**Comments to the Author**

1. Does the manuscript provide a valid rationale for the proposed study, with clearly identified and justified research questions?

Reviewer #1: Partly

Reviewer #2: Partly

2. Is the protocol technically sound and planned in a manner that will lead to a meaningful outcome and allow testing the stated hypotheses?

Reviewer #1: Yes

Reviewer #2: Partly

3. Is the methodology feasible and described in sufficient detail to allow the work to be replicable?

Reviewer #1: Yes

Reviewer #2: Yes

4. Have the authors described where all data underlying the findings will be made available when the study is complete?

Reviewer #1: Yes

Reviewer #2: No

5. Is the manuscript presented in an intelligible fashion and written in standard English?

Reviewer #1: No

Reviewer #2: No

6. Review Comments to the Author

You may also provide optional suggestions and comments to authors that they might find helpful in planning their study.

Reviewer #1: I'm not qualifyed to evaluate a quality of standard English, because I'm not native English spaker.

Reviewer #2: Thank you for your responses to my previous comments.

I am unclear about some of the changes here - the dataset was previously to be made available upon application - now it is to uploaded to a website (the sentence here still needs attention). Clinicaltrials.gov will accept the results, but what about the underlying individual patient data?

The English generally still needs considerable attention, and the literature search still talks about such things as "fewer literatures" - what does this mean? What level of evidence is there here? Have there been any studies including patients with thoracoscopic surgery? What is current standard of care here?

The authors I think misunderstand the concept of uncertainty - this is another word for equipoise - in other words, are people genuinely uncertain about what to do in this situation - this needs to be evidenced in terms of current practice and why the results leading to approval cannot be extrapolated.

For the sample size it needs to be specified that there will be 7 measurements per individual; what allowance for missing data is given in terms of not having all 7 in some cases? One of these is not an outcoem but a baseline measure as it is before induction so there are only 6 outcome timepoints here.

The words shedding rate are still in the paper (line 1457 in the marked up copy).

Formatting of table 1 makes it look like the monitoring is in one arm only - please centralise the things to be done in both groups.

Section 2.7.1 seems to have redundancy about time points (there is also a "the" missing).

The literature gives a long range in terms of duration of the operation - how are operations lasting less than 2 hours to be dealt with in terms of time points?

Thank you for clarifying 2.10.3 - presumably 2.10.4 really means that if one arm reaches 63 patients then the remaining patients will draw the other arm? As it stands it sounds like the trial will go on until there are equal numbers in each arm irrespective of the number required to get there.

Are the opaque envelopes to b prepared in advance or on the day? What safeguards are there to ensure that there is no foreknowledge?

7. PLOS authors have the option to publish the peer review history of their article (what does this mean?). If published, this will include your full peer review and any attached files.

Reviewer #1: No

Reviewer #2: No

---

## [Author Response · Author response to Decision Letter 1]

8 Apr 2024

Dear Editor and Reviewers:

Thank you for your letter and for the reviewers’ comments concerning our manuscript entitled “Effects of ciprofol infusion on hemodynamics during induction and maintenance of anaesthesia and on postoperative recovery in patients undergoing thoracoscopic lobectomy: study protocol for a randomized, controlled trial”.

(Manuscript ID: PONE-D-23-32710R1).

Those comments are all valuable and very helpful for revising and improving our manuscript, as well as the important guiding significance to our researches. We have studied comments carefully and have made correction which we hope meet with approval. The reviewers’ comments are laid out below in italicized font and specific concerns have been numbered. Our response is given in normal font.

Revised portion are marked in red in the paper. 

Reviewer 1

Response to reviewer 1 : Thank you for your rigorous consideration once again. 

We invited professional native English speakers to guide and carefully revise our manuscript.-----And here we did not list the changes but marked in red in revised paper. Once again, thank you very much for your comments and suggestions.

Reviewer 2

1.Comment: I am unclear about some of the changes here - the dataset was previously to be made available upon application - now it is to uploaded to a website (the sentence here still needs attention). Clinicaltrials.gov will accept the results, but what about the underlying individual patient data?

Response: We thank the reviewer for pointing this out. We have removed the original description and replaced it with “No datasets were generated or analysed during the current study now. Data supporting this study will be available from https://figshare.com/ at DOI:10.6084/m9.figshare.25421503, when the clinical trial is completed.” (Page 2,Line 27-29)

2. Comment: The English generally still needs considerable attention, and the literature search still talks about such things as "fewer literatures" - what does this mean? What level of evidence is there here? Have there been any studies including patients with thoracoscopic surgery? What is current standard of care here?

Response: We apologize for our errors. We thank the reviewer for pointing this out. We have change “fewer literatures” to “few studies”. (Page 4,Line 82). A series of clinical studies have been conducted to evaluate the sedative effect of ciprofol in various procedures and settings, including gastroscopy and colonoscopy, fiber-optic bronchoscopy[1, 2], general anesthesia in elective surgeries, and mechanical ventilation in intensive care units[3]. Some studies of ciprofol for anesthesia induction did not clearly point out whether these studies included thoracic surgery patients [4] [5].Other studies on the induction of ciprofol for elective surgery specifically excluded thoracic surgery patient [6-11]. For all we know, although part of the studies on the induction of ciprofol included thoracic surgery patients, there were no studies of ciprofol for anesthesia maintenance in thoracic surgery.

1. Wu B, Zhu W, Wang Q, Ren C, Wang L, Xie G. Efficacy and safety of ciprofol-remifentanil versus propofol-remifentanil during fiberoptic bronchoscopy: A prospective, randomized, double-blind, non-inferiority trial. Front Pharmacol. 2022;13:1091579. Epub 2023/01/10. doi: 10.3389/fphar.2022.1091579. PubMed PMID: 36618929; PubMed Central PMCID: PMCPMC9812563.

2. Luo Z, Tu H, Zhang X, Wang X, Ouyang W, Wei X, et al. Efficacy and Safety of HSK3486 for Anesthesia/Sedation in Patients Undergoing Fiberoptic Bronchoscopy: A Multicenter, Double-Blind, Propofol-Controlled, Randomized, Phase 3 Study. CNS Drugs. 2022;36(3):301-13. Epub 2022/02/15. doi: 10.1007/s40263-021-00890-1. PubMed PMID: 35157236; PubMed Central PMCID: PMCPMC8927014 have no conflict of interest.

3. Lu M, Liu J, Wu X, Zhang Z. Ciprofol: A Novel Alternative to Propofol in Clinical Intravenous Anesthesia? Biomed Res Int. 2023;2023:7443226. Epub 2023/01/31. doi: 10.1155/2023/7443226. PubMed PMID: 36714027; PubMed Central PMCID: PMCPMC9879693 personal relationships that could have appeared to influence the work reported in this paper.

4. Duan G, Lan H, Shan W, Wu Y, Xu Q, Dong X, et al. Clinical effect of different doses of ciprofol for induction of general anesthesia in elderly patients: A randomized, controlled trial. Pharmacol Res Perspect. 2023;11(2):e01066. Epub 2023/02/23. doi: 10.1002/prp2.1066. PubMed PMID: 36811327; PubMed Central PMCID: PMCPMC9944862.

5. Ding YY, Long YQ, Yang HT, Zhuang K, Ji FH, Peng K. Efficacy and safety of ciprofol for general anaesthesia induction in elderly patients undergoing major noncardiac surgery: A randomised controlled pilot trial. Eur J Anaesthesiol. 2022;39(12):960-3. doi: 10.1097/eja.0000000000001759. PubMed PMID: 36214498.

6. Wang X, Wang X, Liu J, Zuo YX, Zhu QM, Wei XC, et al. Effects of ciprofol for the induction of general anesthesia in patients scheduled for elective surgery compared to propofol: a phase 3, multicenter, randomized, double-blind, comparative study. Eur Rev Med Pharmacol Sci. 2022;26(5):1607-17. Epub 2022/03/19. doi: 10.26355/eurrev_202203_28228. PubMed PMID: 35302207.

7. Zhu Q, Luo Z, Wang X, Wang D, Li J, Wei X, et al. Efficacy and safety of ciprofol versus propofol for the induction of anesthesia in adult patients: a multicenter phase 2a clinical trial. Int J Clin Pharm. 2023;45(2):473-82. Epub 2023/01/22. doi: 10.1007/s11096-022-01529-x. PubMed PMID: 36680620; PubMed Central PMCID: PMCPMC10147789.

8. Liang P, Dai M, Wang X, Wang D, Yang M, Lin X, et al. Efficacy and safety of HSK3486 vs. propofol for the induction and maintenance of general anaesthesia: A multicentre, single-blind, randomised, parallel-group, phase 3 clinical trial. Eur J Anaesthesiol. 2023. Epub 2023/01/18. doi: 10.1097/eja.0000000000001799. PubMed PMID: 36647565.

9. Chen BZ, Yin XY, Jiang LH, Liu JH, Shi YY, Yuan BY. The efficacy and safety of ciprofol use for the induction of general anesthesia in patients undergoing gynecological surgery: a prospective randomized controlled study. BMC Anesthesiol. 2022;22(1):245. Epub 2022/08/04. doi: 10.1186/s12871-022-01782-7. PubMed PMID: 35922771; PubMed Central PMCID: PMCPMC9347095.

10. Qin K, Qin WY, Ming SP, Ma XF, Du XK. Effect of ciprofol on induction and maintenance of general anesthesia in patients undergoing kidney transplantation. Eur Rev Med Pharmacol Sci. 2022;26(14):5063-71. Epub 2022/08/03. doi: 10.26355/eurrev_202207_29292. PubMed PMID: 35916802.

11. Akhtar SMM, Fareed A, Ali M, Khan MS, Ali A, Mumtaz M, et al. Efficacy and safety of Ciprofol compared with Propofol during general anesthesia induction: A systematic review and meta-analysis of randomized controlled trials (RCT). J Clin Anesth. 2024;94:111425. Epub 2024/02/28. doi: 10.1016/j.jclinane.2024.111425. PubMed PMID: 38412619.

12. Ozdogan HK, Cetinkunar S, Karateke F, Cetinalp S, Celik M, Ozyazici S. The effects of sevoflurane and desflurane on the hemodynamics and respiratory functions in laparoscopic sleeve gastrectomy. J Clin Anesth. 2016;35:441-5. Epub 2016/11/23. doi: 10.1016/j.jclinane.2016.08.028. PubMed PMID: 27871572.

3. Comment: The authors I think misunderstand the concept of uncertainty - this is another word for equipoise - in other words, are people genuinely uncertain about what to do in this situation - this needs to be evidenced in terms of current practice and why the results leading to approval cannot be extrapolated.

Response: We totally understand the reviewer’s concern. Effects of ciprofol infusion on hemodynamics during induction and maintenance of anaesthesia and on postoperative recovery in patients undergoing thoracoscopic lobectomy are still unclear. According to previous studies, compared with propofol, ciprofol has less effect on circulation and respiration, and its application in general anesthesia sedation is not inferior to propofol. Now we are collecting data and haven't analysed the data yet, the results of the study is uncertain. As the reviewer mentioned,we can't conclude that ciprofol is a better choice of anesthetic agent compared with propofol. 

We added the paragraph “It is not yet known whether ciprofol can prevent the stress response caused by surgery, provide sufficient sedation and maintain hemodynamic stability during thoracoscopic lobectomy.” 

We deleted this paragraph “Conversely , because less effect on circulation and respiratory of ciprofol,quality of recovery in patients used ciprofol maybe is better than that of patients used propofol”. We revised as follow: “Moreover, because less effect on circulation and respiratory, fewer injection pain, ciprofol may be a promising alternative general anesthetics to propofol.” We have also added several more recent references. (page 19-20) 

And here we did not list all the changes but marked in red in revised paper.

Thank you for your rigorous consideration once again.

4.Comment: For the sample size it needs to be specified that there will be 7 measurements per individual; what allowance for missing data is given in terms of not having all 7 in some cases? One of these is not an outcoem but a baseline measure as it is before induction so there are only 6 outcome timepoints here.

Response: Thank you for your valuable comments. We totally understand the reviewer’s concern. Our data will be collected by independent researchers. The compliance of the participants is good. The data missing rate is low. If there is a missing value, we will use the method of multiple imputation for processing and conduct sensitivity analysis. 

-------In section of 2.11 data management ,we have added the following content “Data will be collected by independent researchers using Case Report Form(CRF). If there is a missing value, we will use the method of multiple imputation for processing and conduct sensitivity analysis.” (Line329-331,page 17) 

We hope that the correction will meet with approval. Thank you once again.

5.Comment: The words shedding rate are still in the paper (line 1457 in the marked up copy).

Response: We are very sorry for not writing clearly. We have changed the “shedding rate” to “drop out rate”.(Line 127, page7) We apologize for our errors. We thank the reviewer for pointing this out.

6.Comment: Formatting of table 1 makes it look like the monitoring is in one arm only - please centralise the things to be done in both groups.

Response: Thank you for your valuable comments. We have centralised the things to be done in both groups. (Page 12 ,Table 1)

7.Comment: Section 2.7.1 seems to have redundancy about time points (there is also a "the" missing).

Response: Thank you for your rigorous consideration. Referring to previous research [12], we delete the time points( such as : 60 minutes after the surgery begins,90 minutes after the surgery begins,120 minutes after the surgery begins). 

Because our purpose is to study the hemodynamic changes during anesthesia induction and maintenance, in order to let the reader more clearly understand the setting of time points, we described the time points separately during induction and maintenance. 

------We revised as follows： MAP and HR will be recorded during anesthesia induction，including the following time points: before induction (T0), after induction（T1 BIS≤60),immediately before intubation (T2),immediately after intubation(T3), 1,3, and 5 minutes after intubation(T4 ,T5,T6). (Line 235-241,page 12-13).

“MAP and HR will be recorded during anesthesia maintenance, including the following time points: when the patient is placed in the lateral decubitus position (OR0),when fiberoptic bronchoscope using for localization at the second time (OR1), immediately before the surgery (OR2), 1,3,5,and 30 minutes after surgical incision and at the end of surgery (OR3,OR4 ,OR5 ,OR6, OR7).”

(Line 242-246,page 13)

We hope that the correction will meet with approval. Thank you once again.

12. Ozdogan HK, Cetinkunar S, Karateke F, Cetinalp S, Celik M, Ozyazici S. The effects of sevoflurane and desflurane on the hemodynamics and respiratory functions in laparoscopic sleeve gastrectomy. J Clin Anesth. 2016;35:441-5. Epub 2016/11/23. doi: 10.1016/j.jclinane.2016.08.028. PubMed PMID: 27871572.

8.Comment: The literature gives a long range in terms of duration of the operation - how are operations lasting less than 2 hours to be dealt with in terms of time points?

Response: Thank you for your valuable comments. We totally understand the reviewer’s concern. Referring to previous research [12], we carefully reconsidered the design of our study and deleted the time points such as “60 minutes after the surgery begins,90 minutes after the surgery begins,120 minutes after the surgery begins”. And we added an additional item in withdrawl criteria “the operation takes less than 30 minutes” , although in our research center, the surgery of thoracoscopic lobectomy lasts at least 30 minutes.(line142-144,page7)

----We revised as follow: “MAP and HR will be recorded during anesthesia maintenance, including the following time points: when the patient is placed in the lateral decubitus position (OR0),when fiberoptic bronchoscope using for localization at the second time (OR1), immediately before the surgery (OR2), 1,3,5,and 30 minutes after surgical incision and at the end of surgery (OR3,OR4 ,OR5 ,OR6, OR7).”

(Line 242-246,page 13)

12. Ozdogan HK, Cetinkunar S, Karateke F, Cetinalp S, Celik M, Ozyazici S. The effects of sevoflurane and desflurane on the hemodynamics and respiratory functions in laparoscopic sleeve gastrectomy. J Clin Anesth. 2016;35:441-5. Epub 2016/11/23. doi: 10.1016/j.jclinane.2016.08.028. PubMed PMID: 27871572.

9.Comment: Thank you for clarifying 2.10.3 - presumably 2.10.4 really means that if one arm reaches 63 patients then the remaining patients will draw the other arm? As it stands it sounds like the trial will go on until there are equal numbers in each arm irrespective of the number required to get there.

Response: Thank you for your valuable comments. We apologize for our errors.

We have deleted this paragraph “If the number of cases in the two groups is not equal, the extra patients in one group can continue to receive random numbers until the two groups are equal.” ---We have revised as follow “If one arm reaches 63 patients ,the remaining patients will continue to receive random numbers until both groups reach at least 63 cases.”(line315-318,page 16)

10.Comment: Are the opaque envelopes to b prepared in advance or on the day? What safeguards are there to ensure that there is no foreknowledge?

 Response: We thank the reviewer for pointing this out. We are very sorry for not writing clearly. This is a single-blind study. To ensure the implementation of blind method , the opaque envelopes will be prepared in advance and handed over to the nurse who are responsible for preparing medications. The chief anesthesiologist of the patient and the nurse will open the sealed bag together on the day of surgery.

We revised as follows: “The generated random allocation will be placed in an opaque sealed bag in advance and handed over to the nurse who are responsible for preparing medications.The chief anesthesiologist of the patient and the nurse will open the sealed bag together on the day of surgery.”(Line 320-322,page 17)

----We appreciate for Reviewers’ warm work earnestly, and hope that the correction will meet with approval.

We have studied comments carefully and have made correction which we hope meet with approval.

We would like to thank the reviewers again for taking the time to review our manuscript.

Best Regards 

Na Guo and Xiumei Song

---

## [Decision Letter · Decision Letter 2]

30 Apr 2024

PONE-D-23-32710R2Effects of Ciprofol Infusion on Hemodynamics during Induction and Maintenance of anaesthesia and on Postoperative Recovery in Patients Undergoing Thoracoscopic Lobectomy: Study protocol for a Randomized, Controlled Trial.PLOS ONE

Dear Dr. Xiumei Song,

Thank you for submitting your manuscript to PLOS ONE. After careful consideration, we feel that it has merit but does not fully meet PLOS ONE’s publication criteria as it currently stands. Therefore, we invite you to submit a revised version of the manuscript that addresses the points raised during the review process.

**ACADEMIC EDITOR: **please carefully assess all the reviewers comments

We look forward to receiving your revised manuscript.

Kind regards,

Silvia Fiorelli

Academic Editor

PLOS ONE

Journal Requirements:

Reviewers' comments:

Reviewer's Responses to Questions

**Comments to the Author**

1. Does the manuscript provide a valid rationale for the proposed study, with clearly identified and justified research questions?

Reviewer #1: Yes

Reviewer #2: Yes

2. Is the protocol technically sound and planned in a manner that will lead to a meaningful outcome and allow testing the stated hypotheses?

Reviewer #1: Yes

Reviewer #2: Partly

3. Is the methodology feasible and described in sufficient detail to allow the work to be replicable?

Reviewer #1: Yes

Reviewer #2: Yes

4. Have the authors described where all data underlying the findings will be made available when the study is complete?

Reviewer #1: Yes

Reviewer #2: Yes

5. Is the manuscript presented in an intelligible fashion and written in standard English?

Reviewer #1: Yes

Reviewer #2: Yes

6. Review Comments to the Author

You may also provide optional suggestions and comments to authors that they might find helpful in planning their study.

Reviewer #1: Minor language polishing is still needed. There are several mistakes with punctuation, a few other minor mistakes like no spaces, irregular plural

- ciprofol may be a promising alternative general anesthetics to propofol[

Reviewer #2: Thank you for your response.

The sample size still does not say how many observations per patient the sample size is based upon - as such the sample size cannot be reconstructed. Please give all the details fed into PASS for repeated measures analysis.

The opaque envelope approach is subject to subversion by opening envelopes out of order, or opening multiple envelopes and allocating out of order. There is an extensive literature on this and there needs to be evidence of how this is going to be policed - e.g. ensuring that envelopes are not opened until consent is given and that the consenting patient is actually treated according to allocation. At present one cannot guarantee robustness of the randomisation process.

7. PLOS authors have the option to publish the peer review history of their article (what does this mean?). If published, this will include your full peer review and any attached files.

Reviewer #1: No

Reviewer #2: No

---

## [Author Response · Author response to Decision Letter 2]

14 May 2024

Dear Editor and Reviewers:

Thank you for your letter and for the reviewers’ comments concerning our manuscript entitled “Effects of ciprofol infusion on hemodynamics during induction and maintenance of anesthesia and on postoperative recovery in patients undergoing thoracoscopic lobectomy: study protocol for a randomized, controlled trial”.

(Manuscript ID: PONE-D-23-32710R2).

Those comments are all valuable and very helpful for revising and improving our manuscript, as well as the important guiding significance to our researches. We have studied comments carefully and have made correction which we hope meet with approval. The reviewers’ comments are laid out below in italicized font and specific concerns have been numbered. Our response is given in normal font.

Revised portion are marked in red in the paper. 

Journal Requirements:

Response：Thank you for your valuable comments. We have reviewed all the references to make sure that all the references are complete and correct. We also retrieved all the literatures in PubMed again. All the literatures that we cited have not been retracted. Thank you for your valuable suggestion again.

Reviewer 1

Reviewer #1: Minor language polishing is still needed. There are several mistakes with punctuation, a few other minor mistakes like no spaces, irregular plural

- ciprofol may be a promising alternative general anesthetics to propofol[

Response to reviewer 1 : We thank the reviewer for pointing this out. We are very sorry for our errors. We have carefully reviewed and revised the whole article. Such as add spaces, check the punctuation, singular, plural, and spelling errors. 

We revised as follow

“also the effect on quality of recovery is still unknown” (page 18, Line361) ( spelling errors)

“ciprofol may be a promising alternative general anesthetic to propofol” (singular, plural errors) (page19,Line 373) 

“We conduct the trial to investigate the effects of ciprofol on perioperative hemodynamics and on quality of recovery from anesthesia in patients undergoing thoracoscopic lobectomy.” (page19,Line 375). (spelling errors)

--And here we did not list the changes but marked in red in revised paper. Once again, thank you very much for your comments and suggestions.

Reviewer 2

Reviewer #2: Thank you for your response.

Comment: The sample size still does not say how many observations per patient the sample size is based upon - as such the sample size cannot be reconstructed. Please give all the details fed into PASS for repeated measures analysis.

Response: Thank you for your valuable comments. We apologize for our errors. We seriously consulted the teachers of statistics at the center and rethought our sample size calculation process. We used a Repeated Measures analysis of variance (ANOVA) to calculate the sample size in the PASS 2021(v21.0.3). We observed 15 measurements of MAP ( T0-6 and OR0-7) per patient during induction and maintenance of anesthesia between the two groups. 

We revised as follow:

“We used a Repeated Measures analysis of variance (ANOVA) to calculate the sample size in the PASS 2021(v21.0.3). According to the results of the preliminary experiment, We observed 15 measurements of MAP ( T0-6 and OR0-7) per patient during induction and maintenance of anesthesia between the two groups. The standard deviation of each group is approximately 13, and the autocorrelation coefficient between the adjacent measurement points of the same study participant is 0.5, the difference of the MAP between the two groups is 3. Type I error is 5% and the power is 80% in the pre-test. In the bilateral test, the sample size of the two groups is equal, and the sample size of each group is 54. Considering a 20% drop out rate, a sample size of 68 patients in each group is required. Thus, the final sample size is 136.”(Page 6-7, Line124-131)

We added the calculation process of the sample size in the supplementary materials. 

We need to explain the reasons for the original calculation of sample size. We choose 7 measurements of MAP (T0-6) during induction of anesthesia to calculate the sample size at first, because our results of the preliminary experiment were consistent with those previously reported, and there were few studies about ciprofol for anesthesia maintenance.

We added the original calculation process of the sample size in the supplementary materials, too.

We choose 15 measurements during induction and maintenance of anesthesia and calculate the sample size again, the current method is more rigorous (the final sample size is 136). Because the primary outcome that will be assessed in this study is the perioperative hemodynamic fluctuation (during induction and maintenance). 

Fortunately, our trial is in the data collection stage, and we are grateful for the valuable suggestions and feedback from the reviewers, which are very helpful for us to improve our study. We hope the correction will meet with approval.

Comment: The opaque envelope approach is subject to subversion by opening envelopes out of order, or opening multiple envelopes and allocating out of order. There is an extensive literature on this and there needs to be evidence of how this is going to be policed - e.g. ensuring that envelopes are not opened until consent is given and that the consenting patient is actually treated according to allocation. At present one cannot guarantee robustness of the randomisation process.

Response: We thank the reviewer for pointing this out. We delete this paragraph “The chief anesthesiologist of the patient and the nurse will open the sealed bag together on the day of surgery. ”

We revised as follow: “If the patient meets the inclusion and exclusion criteria, and has signed the informed consent, the anesthesiologist of the patient and the nurse will open the opaque sealed bag sequentially on the day of surgery and the consenting patient is treated according to allocation.” (page 15, Line 298-305)

----We appreciate for Reviewers’ warm work earnestly, and hope that the correction will meet with approval.

We have studied comments carefully and have made correction which we hope meet with approval.

We would like to thank the reviewers again for taking the time to review our manuscript.

Best Regards 

Na Guo and Xiumei Song

---

## [Decision Letter · Decision Letter 3]

31 May 2024

Effects of Ciprofol Infusion on Hemodynamics during Induction and Maintenance of anesthesia and on Postoperative Recovery in Patients Undergoing Thoracoscopic Lobectomy: Study protocol for a Randomized, Controlled Trial.

PONE-D-23-32710R3

Dear Dr. Xiumei Song,

We’re pleased to inform you that your manuscript has been judged scientifically suitable for publication and will be formally accepted for publication once it meets all outstanding technical requirements.

Kind regards,

Silvia Fiorelli

Academic Editor

PLOS ONE

Additional Editor Comments (optional):

Congratulations to the authors and thanks to the reviewers for the provided suggestions which really helped improve the quality of the manuscript

Reviewers' comments:

Reviewer's Responses to Questions

**Comments to the Author**

1. Does the manuscript provide a valid rationale for the proposed study, with clearly identified and justified research questions?

Reviewer #1: Yes

Reviewer #2: Yes

2. Is the protocol technically sound and planned in a manner that will lead to a meaningful outcome and allow testing the stated hypotheses?

Reviewer #1: Yes

Reviewer #2: Yes

3. Is the methodology feasible and described in sufficient detail to allow the work to be replicable?

Reviewer #1: Yes

Reviewer #2: Yes

4. Have the authors described where all data underlying the findings will be made available when the study is complete?

Reviewer #1: Yes

Reviewer #2: Yes

5. Is the manuscript presented in an intelligible fashion and written in standard English?

Reviewer #1: Yes

Reviewer #2: Yes

6. Review Comments to the Author

You may also provide optional suggestions and comments to authors that they might find helpful in planning their study.

Reviewer #1: There are still some language issues in the manuscript that need to be corrected, although I assume that the publisher has a final check on each manuscript. If so, no new audit is required.

For example "..clinical trial participations were patients...".

Reviewer #2: thank your for addressing my prior comments - i have nothing further to add to my previous thoughts on this protocol

7. PLOS authors have the option to publish the peer review history of their article (what does this mean?). If published, this will include your full peer review and any attached files.

Reviewer #1: No

Reviewer #2: No
